# How to Use Questionnaire Results in Psychosocial Risk Assessment: Calculating Risks for Health Impairment in Psychosocial Work Risk Assessment

**DOI:** 10.3390/ijerph18137107

**Published:** 2021-07-02

**Authors:** Jan Dettmers, Christiane R. Stempel

**Affiliations:** Department of Work and Organizational Psychology, Faculty of Psychology, University of Hagen, 58084 Hagen, Germany; christiane.stempel@fernuni-hagen.de

**Keywords:** psychosocial risk assessment, cutoff values, job stressors, job resources, questionnaires, psychological health impairment

## Abstract

Psychosocial risk questionnaires are common instruments in occupational safety and health promotion. Organizations use psychosocial risk questionnaires to obtain an economic overview of psychological job stressors and job resources. However, the procedures to assess if a result for a given workplace group is critical and calls for further action differ significantly and are often based on an arbitrary rule of thumb instead of empirically based evaluations. This article presents a method to translate questionnaire results into risk values for the occurrence of health impairment. We test this method on a dataset including the job stressors, job resources, and emotional exhaustion of 4210 employees from different industries. We applied logistic regression analysis to calculate the risks for impaired psychological health, indicated by high values of the burnout indicator emotional exhaustion. The results indicate significantly different health impairment risks (probabilities) for different scores on the job stressors and job resources scales as well as for scale score combinations. The risk values can be used to define cutoff values between high- and low-risk workplaces that are empirically based on stressor–strain relationships and are easily understandable by all stakeholders in the psychosocial risk assessment process, including laypersons.

## 1. Introduction

Psychosocial work risk questionnaires are standard tools in occupational health and safety promotion, in general, and work risk assessment, in particular. Organizations use the results of standardized written surveys to obtain a quick and economic overview of psychological job stressors and resources. The questionnaire results help identify psychological hazards and select specific workplace groups that should receive particular attention for further action [1]. However, unlike risk assessment results regarding physical or chemical hazards, the results of psychosocial risk questionnaires often lack meaningful cutoff values or empirically based indications on how to interpret specific scores. Many psychological measures (e.g., Copenhagen Psychosocial Risk Questionnaire COPSOQ II [2]) provide statistical reference values based on extensive and industry-specific samples. However, these references do not indicate if a specific value is good, acceptable, or bad regarding the specific health impairment risk. At most, they provide a comparison with an industry-specific status quo that still might represent unacceptable working conditions for psychological health and well-being. However, practitioners need clear indications of how to interpret the results of questionnaire surveys in terms of actual health risks to make conclusions about the urgency of taking measures within a concerned workplace group.

This paper presents a method for calculating risk values for the occurrence of health impairment from psychosocial risk questionnaire scores. The method is tested on an exemplary dataset that includes the job stressors, job resources, and indicators of psychological health of 4210 employees from different industries. Using emotional exhaustion as an indicator of impaired psychological health, we applied logistic regression analysis to calculate the risks for impaired psychological health for given scores or score combinations of job stressors and/or job resources. The focus is not on replicating findings on the general effects of psychological job stressors and job resources on strain but on proposing an approach to translate expressions of job stressors and job resources identified with a specific psychosocial risk questionnaire into probabilities (risks) for psychological health impairment. Instead of the (often arbitrary) rules of thumb for questionnaire result interpretations, we wish to establish an assessment procedure based on underlying empirical stressor–strain relationships.

Thus, with this paper, we further professionalize the practice of psychosocial work risk assessment. Particularly, we aim to provide concrete guidance to practitioners on how to interpret questionnaire results in a way that is comprehensible to all stakeholders involved in the risk assessment process. The proposed method is generic and not limited to specific questionnaires or outcomes. It can be applied to any valid and reliable psychosocial risk questionnaire but also to various indicators of psychological well-being and (ill-) health. Thus, our study contributes to creating an accessible scientific and methodological basis for the practical challenges of psychological work risk assessment.

### 1.1. Psychosocial Work Risk Assessment

In many Western industrial countries, companies have a legal obligation to undertake risk assessments for all workplaces and activities to preserve employee health and to prevent accidents or other harmful work effects [3]. Traditionally, work risk assessment focused on risks for accidents and physical hazards, such as excessive noise, heat, or chemical hazards. However, in recent years the assessment of psychological hazards has gained attention, as research has accumulated knowledge regarding the effects of psychosocial job stressors and job resources on employee health [3,4]. The respective research findings are reflected in the legislative regulation for health and safety, including the obligation to account for psychological hazards in work risk assessment (European Occupational Safety and Health Framework Directive, 89/391/EE).

As in physical risk assessment, psychosocial work risk assessment focuses on objective, generalizable workplace hazards rather than on individual factors. Examples are psychological stressors and resources regarding work content (e.g., emotional demands), work organization (e.g., work intensity), or social relationships (e.g., social support). A significant difference between physical work risk assessment and psychological work risk assessment regards the required integration of the employees into the assessment process and the applied tools. Even though the concerned psychological stressors in psychosocial risk assessment derive from objective characteristics of the work situation, assessing the stressors requires the participation and subjective perception of the workers themselves. Therefore, in most cases, subjective assessment tools, such as questionnaires, workshops, or interviews (often combined with observations), are typical tools within psychosocial work risk assessment.

Another typical feature of the psychosocial risk assessment process is the stepwise proceeding from general to specific [5,6]. For example, the risk assessment process may start with a general orientation on potential psychosocial hazards based on general reports from experts and responsible persons within the organization, followed by a more systematic overview using screening instruments such as questionnaires. Finally, in-depth analysis and more focused measures (e.g., workshops or systematic observations) are applied to derive and evaluate actions that address the identified psychosocial hazards. The results of screening instruments such as standardized questionnaires are mostly too abstract to derive concrete work redesign measures that may reduce the psychosocial hazards. That is why further analyses (e.g., workshops) that follow the written survey results are required [5]. However, these instruments require more time and resources and often cannot be applied to the general population of workplaces within the organization. Thus, the objective of the applied screening instrument is to distinguish workplaces with hazards that constitute a low health impairment risk from workplaces with hazards that constitute a high health impairment risk for employee health in order to direct efforts and resources required by further measures to the most urgent targets.

### 1.2. Prioritization and the Role of Questionnaire Results

For screening the general situation of psychosocial hazards in the organization, questionnaires are standard (see [7] for an overview). Standardized surveys provide a quick and economic overview of psychological job stressors and job resources. The results can be structured and differentiated between workplace groups or work teams. Apart from the basic overview, the results of valid and reliable questionnaires have the potential to distinguish workplace groups that face particular hazards and have to be addressed with a high priority from other workplace groups facing lower psychological hazards. However, how the questionnaire results indicate that the measured psychological hazards are critical is not trivial. In some cases, practitioners and experts of the risk assessment process refrain from the solely quantitative interpretation of questionnaire results and rather take them as additional information in the discursive decision process in steering groups or other management instances [5]. Here, the survey provides confirmatory results that must be integrated and combined with existing background information. In other cases, quantitative rules are defined on how to interpret the questionnaire results and which numerical expressions of questionnaire scores indicate further action. However, in those cases as well, the applied rules to decide whether a questionnaire score means a severe hazard or which workplace group to select for further action may differ significantly in practice. In the following, we outline some typical approaches.

### 1.3. Uniform Cutoff Values

Many commonly applied rules are intuitive and arbitrary without any reference to the identified psychosocial work risks. An example of such rules of thumb is uniform cutoff values for the applied questionnaire scales. For example, a specific psychological job stressor, such as work interruptions or time pressure, is measured with items capturing work interruptions, and respondents indicate on a 5-point agreement scale to which degree the featured situations are representative for their work situation. The “uniform cutoff procedure” then would define a uniform scale score (e.g., “3.5”) for all measured job stressors (e.g., work interruptions, time pressure, social stressors) as a cutoff value indicating a problematic expression of the respective job stressor.

A problem with this method is that the specific scale scores are arbitrarily defined (why not 3.25 or 3.75?) and essentially meaningless to the actual risk for employee health impairment. First, the uniform definition of scale values for different job stressors does not account for the different exposure intensities in the population. Second, it does not account for the varying relationships of different job stressors with employee health impairments. To illustrate the problem, we consider the following scales of the Health and Safety Executive Management Standards (HSE MS) Indicator Tool [8]: job demands (*M* = 2.95, *SD* = 0.15) and (problematic) relationships (*M* = 2.23, *SD* = 0.32), which can take values between 0 and 5. Taking the same cutoff values for both job demands and (problematic) relationships would neglect the differing mean values in the population. For instance, a score of 3 corresponds to a value of more than two standard deviations above the general mean for the relationship scale but is well within one SD for the mean of job demands.

### 1.4. Reference Value-Based Assessment of Questionnaire Results

A method that better addresses the different meanings of identical numerical expressions is the “reference value-based” method. This method applies comparative statistical values when assessing the need for action. Popular psychosocial questionnaires, such as COPSOQ [2] or the HSE MS Indicator Tool [8], feature impressive databases that permit the creation of industry-specific reference values with which the scores of one’s own organization can be compared. The reference value-based assessment approach calls for more in-depth analysis or prioritized processing if significant deviations from the respective industry-specific mean values are identified. For instance, referring to the aforementioned example from the HSE MS Indicator Tool [8] with job demands (*M* = 2.95, *SD* = 0.15) and relationships (*M* = 2.23, *SD* = 0.32), a questionnaire score of 2.75 in a workplace group on the problematic relationship scale would indicate a higher need for action than a score of 3.00 on the job demands scale, despite the lower absolute value.

Accordingly, the orientation on industry-specific mean values might eliminate the different meanings of the identical questionnaire scores on different work stressor scales within one industry. At the same time, comparisons with industry-specific reference values are not without problems if the need for action is to be defined on their basis. This is because the deviations from the industry-specific mean, which indicate the need for action, are, at best, indirectly related to the actual health risk but do not specify it. The reference value-based approach only depicts statistical realities and establishes a purely statistical norm, which can be problematic, particularly when using industry-specific mean values. For example, we may consider the structure of working hours of healthcare professionals, which may have undoubtedly already problematic mean values within the industry population [9,10]. Under such circumstances, a comparison with inadequate benchmarks may result in no significant deviations from industry mean values and mask the need for action as indicated from an occupational health perspective.

In addition to the potentially inherently problematic reference values, the possibly uniform determination of deviations from this value (e.g., one standard deviation) can be problematic. Determining standardized deviations from a benchmark considers too little the different effect sizes of the potential stress loads. Comparing different effect sizes of stressors by meta-analyses [4,11,12] shows substantial differences in the levels of correlations between different psychological stressors and impaired health. For instance, drawing on the meta-analysis of Fila and colleagues [12], even a standardized deviation of 0.5 in job demands (|rc|¯ = 0.51) would stand for a different effect on emotional exhaustion as the same deviation for lack of control (|rc|¯=0.20).

In summary, the prevalent methods for assessing and interpreting questionnaire data in the context of psychosocial risk assessments are partly based on empirically unsubstantiated assumptions and have solely indirect or low relation to the actual health risk. However, assessing the actual health risk of employees is a major aim of the risk assessment not only of psychological stress but of all hazards at work. We propose that well-constructed psychological questionnaires are valid and reliable measurement instruments that can be used to provide more than just heuristic guidance. Written survey results should be able to provide quantitative conclusions about health risks based on empirical findings, which can then be used to decide how to proceed in the psychosocial risk assessment process.

### 1.5. Risk-Based Interpretation of Questionnaire Results

Work psychological stress research provides sufficient empirical evidence for robust associations between the levels of psychological job stressors and psychological impairment [4,11,13]. The results of longitudinal studies on mental stress and strain data suggest that the relationships can be understood as causal relationships in the sense that high levels of psychological job stressors can contribute to the development or increase of mental health impairment [14]. Stress theoretical models [15] suggest that stressors do not act on passive individuals and that the same stressors may have different effects on different individuals [16]. Individuals take action to cope with stress, and personal characteristics such as optimism or neuroticism moderate the effects of stressors [16,17]. However, even if interindividual differences and the proactive role of individuals in dealing with stressors are considered [18], robust probabilistic associations between stressors and health impairments persist [4]. Psychosocial risk questionnaires may validly and reliably record the levels of psychosocial stressors that employees experience at work [19]. Subjective tendencies in the response behavior may impact the questionnaire results of different individuals confronted with the same stressors. However, inter-rater reliabilities between job incumbents and external observers [6,20] as well as the intraclass correlations between the statements of employees at similar workplaces [19] underline that the questionnaires not only capture subjective impressions but also objectively existing psychosocial stressors. Both the robust correlations between psychosocial stressors and health impairments and the availability of mental stress questionnaire instruments allow probabilistic statements about the risk of health impairments based on specific scores on questionnaire scales. Different approaches have been developed in the past for this purpose.

### 1.6. Ranking of Psychosocial Work Risks

To assess health risks based on questionnaire results, Clarke and Cooper [21] proposed a methodology that assesses the relative risk of individual psychosocial stress factors in a specific workplace group based on empirically found relationships between job stressors and health impairment. This approach focuses on prioritizing stress factors, i.e., deciding which stress factor should be addressed first and with the highest effort. Following Warner’s [22] definition for risk, Clarke and Cooper [21] determine risk as a combination of the probability of the stressor’s occurrence and the amount of damage caused by the stressor. Psychological job stressors act mainly as a chronic impact of the work situation (e.g., daily hassles) rather than as single events. Therefore, Clark and Cooper [21] consider the chronic intensity of a psychological stressor (exposure)—as it is measured by a questionnaire—as the probability in their risk definition. Furthermore, Clarke and Cooper [21] suggest using the correlation between the job stressor and health impairment indicators to quantify the extent of harm or damage caused by the job stressor. For example, a correlation of *r* = 0.40 is found between work intensity and the number of absence days per year in a workplace group. That would mean that work intensity can explain *R*^2^ = 16% of the variance in the number of absence days. According to Clarke and Cooper [21], this variance explanation is used in the logic of the risk matrix as the “extent of damage” (consequences). For calculating the risk factor of a specific work stressor, Clarke and Cooper [21] propose the formula risk factor (R) = exposure (E) × consequences (C). As an example, Clarke and Cooper [21] draw on a study by McFarlane [23] in a UK retail company in which 66 commercial employees completed the Occupational Pressure Management Indicator (PMI) and the General Health Questionnaire GHQ [24]. In their example, the PMI for workload resulted in a score of 9.85 (Exposure E). The correlation between workload and psychological impairment measured by GHQ is *r* = 0.09, corresponding to *R*^2^ = 0.0081 or 0.81%. This results in a risk value associated with the workload stressor for this workgroup of R = E × C = 9.85 × 0.81 = 7.98.

Clarke and Cooper [21] acknowledge that the value of 7.98 is initially not meaningful in itself. The value does not describe a specific unit or an actual probability for developing a stress-related health problem. Rather, it is a matter of comparing and ranking health risks caused by different stressors. In their example, they bring up the different stressors (PMI) measured in the specific example, which have lower scores on the questionnaire scales (in this case, work–family conflict with 8.91) but, due to the higher correlation with health impairments (*r* = 0.16; *R*^2^ × 100 = 2.4) with R = E × C = 8.91 × 2.4 = 21, result in a higher risk value and should therefore be prioritized.

In summary, directly linking psychosocial stressors to the empirically determined correlation of health impairments is advanced compared to the absolute or relative value interpretation of psychosocial work risk questionnaire scales. This makes it possible to prioritize and rank stressors that employees confront within a particular workplace. However, this approach does not provide a meaningful statement about the absolute health risk in a specific workplace. In this respect, no statement can be made as to whether one or more questionnaire scores are acceptable or not, i.e., whether there is a fundamental need for action or not. In addition to the low absolute informative value of the “risk values” determined by the Clarke and Cooper [21] approach, the methodological approach is not without problems. Clarke and Cooper [21] explicitly propose determining C from the correlations within the specific workplace group. However, this correlation could be very low, especially in very high stressor values with slight variance between the individual persons concerned, not because the stressor has no relevance for the outcome, but because the variance restriction would artificially reduce the correlation.

### 1.7. Prioritization of Workplace Groups: Absolute Thresholds of Questionnaire Values for Increased Risk of Health Impairment

While Clarke and Cooper [21] are concerned with comparative assessments of different mental risks, Zeike et al. [25] present an approach to identify cutoff scores on questionnaire scales that indicate a high risk for health impairment. This approach aims to identify the cutoff values of psychosocial job stressors and job resource questionnaire scales that differentiate between high and low risks for psychological health impairments (e.g., depression, burnout). Based on Karasek’s [26] job demand-control model, Zeike et al. [25] refer to the stress-relevant job characteristics demand and control. Using the Job Content Questionnaire [27], they aim to identify those scores for demand and control that are associated with a high probability of having scores on the WHO-5 Well-Being scale [28] that are indicative in depression testing [29].

By dichotomizing the health indicator measured by the WHO-5 scale score into < 13 (mental health impairment) vs. ≥ 13 (no mental health impairment) [29], Zeike et al. [25] aim to find those thresholds on the demand and control scale that best differentiate between individuals who are above or below the critical WHO-5 threshold. They apply ROC (receiver operating characteristic) analysis to optimize the threshold’s sensitivity and specificity by maximizing the Youden index, indicating the best combination of sensitivity and specificity. Based on sufficient correlations between demand and control as predictors and the dichotomous criterion (health impairment yes or no), a score of 31.4 (on a scale of 12 to 48) is identified as the cutoff value on the demand scale that reaches the maximum Youden index. On the control scale, the score of 34.5 reaches the maximum Youden index.

Zeike et al.’s [25] approach using dichotomous outcomes, ROC analysis, and the maximization of the Youden index is typical for determining cutoff values. However, this approach is not without limitations for the practical application of the psychosocial work risk assessment. Even a cutoff value determined by ROC analysis cannot indicate much about whether a score near the cutoff value is acceptable because the risk level itself is not explicitly specified. Furthermore, the maximization of the Youden Index to optimally account for sensitivity and specificity may be indicative for one stress factor, but results regarding sensitivity and specificity change if we consider multiple stress factors. In concrete terms, this means that an approach based on ROC analyses for each stress factor could lead to many false-positive results. The thresholds determined in this way for differentiating between critical and noncritical exposure levels only achieve a rate of 63.3% correct classifications (true positives + true negatives) regarding the occurring health risks [25]. This could undermine the acceptance of the threshold values that have been determined with the approach. An instrument must clearly distinguish between workplaces where urgent action is required and workplaces where simpler procedures can be chosen to decide which workplaces, for example, should have more in-depth analyses and additional resources mobilized. Furthermore, this distinction must be comprehensible for all stakeholders. Therefore, instead of abstract thresholds for questionnaire values by ROC analyses, we propose interpreting questionnaire results in terms of concrete and absolute risk determinations.

### 1.8. Study Aim

The presented approach aims to provide a simple method to translate questionnaire results within psychosocial risk assessment into meaningful information that is understandable by all stakeholders in the risk assessment process, including laypersons. We believe that risks and risk ratios are generally understandable, provide a solid basis for transparent decision making, and are an excellent basis for establishing rules on how to proceed during the different stages of psychosocial risk assessment. Therefore, we propose a method that translates raw scores from psychosocial questionnaires into easily interpretable risk values. To put our method to the test, we draw on a dataset of 4210 employees that completed a psychosocial risk questionnaire and a scale for impaired psychosocial health.

## 2. Methods and Materials

Whereas the objective of linear regression analysis is to predict continuous values of outcome variables based on the scores of one or more predictor variables, the logistic regression analysis aims at predicting the probability of a dichotomous outcome variable. In the case of our study, we wish to predict the risk of health impairment based on the scores of one or more job stressors and job resources scales. In the following, we demonstrate this procedure on an exemplary empirical dataset that includes data on job stressors and job resources and health indicators.

### 2.1. Procedure and Sample

The data collection draws on various sources and single studies using the same questionnaire. A part of the dataset (*n* = 2019) derives from the data collected by an ISO-certified (ISO 26362) German online panel provider Respondi (accessed on 20 April 2021), another part of the dataset was collected within a students’ project at the University of Hagen and multiple master theses (*n* = 2611), and a third part was collected in practical projects of psychosocial risk assessments (*n* = 204). All data collections were online surveys. To fulfill established ethical criteria, all participants were asked to provide their informed consent for participation. Before the actual survey, the first page of the online questionnaire informed participants about the aim of the study and the voluntary nature of participation. No deception about the study aim and no intervention or consulting took place. In the informed consent form, participants had to formally declare that they agreed to participate in the research project, that they understood the aim of the study, that the personal perception of working conditions and individual well-being was assessed, that participation included the risk of perceiving negative emotions, and that they confirmed to accept this risk. Furthermore, participants were informed that study participation was completely voluntary, and discontinuation or interruption of the participation was possible at any point in time without being identified and without any disadvantages. Finally, participants hat to declare that they understood all of the abovementioned issues. On a second page, participants were informed about the data processing procedures in accordance with the European (and German) data protection regulations and about the institutional responsibilities and the data protection officer. Generally, the survey was anonymous, and no personal data were collected that could be used to identify single participants.

We excluded 130 participants that exhibited a conspicuous response pattern on the online survey (swift completion, no variation among items). Furthermore, we included only those participants with regular employment of at least 25 h per week. The final sample consisted of 4210 employees (56% female). The mean age of the employees was 38 years (*SD* = 9 years). A total of 46% of the participants had a university degree. Most participants were employed in service and administrative work (30% commercial and sales employees, 21% administration), a further 21% of the participants were technical or trade employees, 20% were employed in social care and medical services, 5% were school or university teachers, and the final part of 5% worked in logistics and transport. On average, participants worked 39.7 h per week (*SD* = 5.7 h).

### 2.2. Measures

#### 2.2.1. Psychological Job Demands and Job Resources

We used the Questionnaire for Psychosocial Work Risk Assessment (FGBU) [19]. This questionnaire is based on the recommendations of the Joint German Occupational Health and Safety Strategy [1] that proposes psychological job stressors and job resources from the areas of work content, work organization, social relationships, and environmental factors that should be accounted for in occupational risk assessments at work. These job stressors and job resources are comparable to stress factors accounted for in occupational health and safety legislation of other countries and respective psychosocial risk assessments [3]. The questionnaire consists of nineteen 3-item scales and an index of 10 physical stressors. The (in total) 67 items were assessed on a 4-point Likert scale ranging from 1 (not correct at all) to 2 (rather not correct), 3 (rather correct), and 4 (fully correct). Table 1 illustrates the questionnaire scales and the internal consistencies as an indicator of reliability in this study.

#### 2.2.2. Psychological Health Impairment

Psychological health impairment was captured by emotional exhaustion as the primary indicator of burnout [30], which is linked to severe psychological illnesses such as depression and suicide [31]. We assessed emotional exhaustion using seven items of the German version [32] of the emotional exhaustion scale taken from the Maslach Burnout Inventory (MBI), 2nd Edition [33]. One sample item for emotional exhaustion is “I feel burned out from my job”. Participants rated the seven items on a 7-point scale ranging from 1 (“never”) to 7 (“daily”). Cronbach’s α was 0.91 in this study.

A particularity of risk in terms of probability of an event is that the outcome is a dichotomous variable: an accident may be defined as dichotomous variable as it may occur or not. Additionally, a specific health impairment, such as a heart attack or cancer, may or may not occur. However, most indicators of psychological health impairments are measured as continuous variables. In emotional exhaustion, the measures distinguish between more or less exhausted but not a priori between exhausted and not exhausted. To establish a dichotomous criterion for psychological health impairment, we therefore consulted suggestions for thresholds for severe expressions of emotional exhaustion or burnout that are either based on common sense or deviations from statistical norms [34,35,36]. We further conversed with health and safety practitioners to define an intensity threshold for strain symptoms that indicate a severe psychological health impairment. Based on these considerations, we defined the weekly or more frequent occurrences of exhaustion symptoms addressed in the MBI emotional exhaustion scale, such as feeling burned out, feeling frustrated, and feeling used up at the end of the workday, as severe psychological health impairments, which corresponds to a score of 5 on the 7-point response scale. Consequently, we dichotomized the mean scores on the MBI Emotional Exhaustion into scores under 5 (less than weekly) and scores from and above 5 or higher (weekly or more frequent symptoms) and defined the latter as psychological health impairment. The general prevalence of scores from and above 5 was 15% in this sample.

### 2.3. Analysis

For each of the 20 job stressors and job resources of the FGBU [19], we conducted logistic regression analysis with dichotomized psychological health impairment as the dependent variable. We then used the regression coefficient (*B*) and the intercept (*A*) to calculate the probability for psychological health impairment indicated by a score on the emotional exhaustion scale equal to or exceeding 5. We entered the score of each specific work stressor score to calculate the specific risk or probability for psychological health impairment associated with the scale score.
(1)P(Psychological Health Impairment)=Exp (A + scorestressor scale× B) 1+Exp + scorestressor scale×B)

Furthermore, we combined theoretically meaningful combinations of job stressors and job resources (e.g., work intensity and job autonomy [26], emotional demands, and social support [37]) for additional analysis. In these cases, we calculated multiple logistic regression equations, entering both scores and coefficients of the considered job stressors and job resources into the equation.

## 3. Results

Table 2 displays the correlations of the emotional exhaustion scale and all job stressors and job resources assessed with the FGBU. As indicated by the significant correlation coefficients, emotional exhaustion is positively related to the assessed job stressors and negatively related to the assessed job resources. This implies that emotional exhaustion can be predicted by the scores on the psychosocial questionnaire scales.

To calculate the general explaining value of all job stressors and job resource scale scores for the dichotomized psychological health impairment variable, we calculated multiple logistic regression analyses with psychological health impairment as the dependent variable and all job stressors and job resources as predictors. The explained variance estimated by Nagelkerkes *R*^2^ [38] was 0.387 (Cox and Snell *R*^2^ = 0.222; [39]), indicating that a substantial part of the variance of psychological health impairment can be explained by all assessed job stressors and job resources together.

In a further step, following the procedure described in the analysis section, we calculated the regression coefficients for all single job stressors and job resources predicting psychological health impairment. In addition, we used these coefficients to calculate the risk values associated with exemplary questionnaire scores. Table 3 displays the results of these calculations. Figure 1 and Figure 2 depict the calculated relationship between questionnaire score and risk for psychological health impairment graphically.

To account for combinations of job stressors and job resources, we calculated multiple logistic regression analyses, entering specific combinations of job stressors and job resources that are discussed in the work stress literature (e.g., job demand-control [26], job demand-control support [12,37]). Based on these theoretical considerations, we calculated logistic regression analyses with psychological health impairment as the dependent variable with (1) work intensity and autonomy [26] and (2) social and emotional stress and social support from supervisor [12,37] as predictors. Table 4 displays the calculated risk values for specific scale score combinations.

To further illustrate the capacity of the approach to distinguish between high risks and low risks for psychological health impairment, we calculated cross tables for low and high scores of the job stressor and job resources scales and psychological health impairment. As the cutoff value, we used the job stressors or job resources score derived from the logistic regression analysis that was related to a double risk (*p* = 0.30) for psychological health impairment compared to *p* = 0.15 as the general prevalence in the sample (this was not applicable for completeness of tasks, variability, and work interruptions). We then divided the sample into groups below and above these cutoff values and counted the number of cases with and without psychological health impairment. Table 5 displays the frequencies of stressor–health–impairment combinations. We applied the approach proposed by [40] to provide person-oriented effect sizes based on percent correct classifications (PCC) in the way that questionnaire scores from and above the defined stressor cutoff scores (respectively from and below the defined resource cutoff scores) predict psychological health impairment. As illustrated in Table 5 for the listed job stressors and job resources, the total of correct classifications (i.e., true positives + true negatives) was at least 81.85%.

## 4. Discussion

### 4.1. General Discussion

This paper aimed to propose a method to translate questionnaire scores into risk values for psychological health impairment. The overall objective was to make questionnaire scores more understandable for all stakeholders of the psychosocial risk assessment process and to provide them with a valid base for informed decisions on the subsequent actions. Extensive research has provided empirical evidence for linking psychosocial job stressors and job resources with psychological health impairment. Therefore, it is possible to assess the urgency of action by the level of psychosocial job stressors and job resources measured by a valid and reliable psychosocial questionnaire. Based on our experiences in psychosocial risk assessment projects, we believe that concrete risk values in terms of probabilities to become ill are much better indications for practitioners than commonly provided raw values on questionnaire scales, correlations, or linear regression coefficients that are often used in psychological research.

We applied logistic regression analysis—an established and easy method to calculate the probabilities for discrete events given the values of specific predictors—to determine the absolute and relative risk for psychological health impairment based on the questionnaire scores of job stressors and job resources. We gave the quantitative results of psychosocial questionnaires a meaning that is much more concrete than simple raw scores on single questionnaire scales. This enables an easier interpretation of the questionnaire results in psychological work risk assessment and facilitates the subsequent course of action. It is important to note that the presented method is independent of a particular psychosocial questionnaire but can be applied to any reliable and valid questionnaires that measure stressors and resources at work. Furthermore, the approach can be extended to a range of indicators for psychological health (impairment).

For the current paper, we applied the proposed method to a dataset collected from employees that completed a questionnaire assessing a wide variety of psychosocial job stressors and job resources and emotional exhaustion as an indicator of psychological health impairment. Based on references in the literature [34,36] and common-sense estimations of health and safety practitioners, we used the cutoff value of 5 (weekly exhaustion symptoms) to dichotomize the psychological health indicator emotional exhaustion into impaired psychological health or no impaired psychological health. Using logistic regression analysis, we found that the assessed psychological job stressors and job resources are related to the risk (in terms of probability) for the occurrence of impaired psychological health. Higher values on psychological job stressors and lower values on psychological job resources are associated with increased risk for impaired psychological health. While this finding is not particularly new or surprising in occupational health psychology [11], we calculate concrete and quantifiable risks, providing a more accessible and comprehensible approach to interpret the questionnaire results. For instance, our method allows us to estimate that, e.g., a certain score of an employee on a psychological risk questionnaire scale means a doubled risk of showing frequent symptoms of emotional exhaustion.

Taking a closer look at our study results, we find that certain psychological stressors (work intensity, social stressors) have a stronger effect regarding increased risk for health impairment than others (e.g., underutilization of skills). Whereas questionnaire scores of 2.8 for social stressors from colleagues already double the risk for health impairment, a maximum score of 4 for underutilization of skills only increases the risk for health impairment by 50%. These differences regarding the impact of stressors must be accounted for when assessing psychological hazards at work.

Furthermore, we found that combinations of job stressors and job resources may change the relationship of the single job stressor and job resource scores on psychological health impairments. For example, with low values on job autonomy (score = 1), work intensity scores of 2.7 may already double the risk for psychological health impairment (*p* = 0.30), whereas with high values of job autonomy (score = 4) the score for work intensity must be as high as 4 to be related to a doubled risk for health impairment. Addressing a persistent deficit of established practices in psychosocial risk assessment, we account for stressor/resource combinations that have long been discussed and empirically tested in occupational stress research [26,41]. Our results suggest that taking job resources into account to predict the impact of a stressor on psychological health impairment should be incorporated into the application of psychosocial risk assessment.

### 4.2. Practical Implications

The aim of our study was not to replicate established findings on the general effects of psychological stressors on ill-health but to provide a method that can be applied in psychosocial risk assessment to improve decisions and results in terms of focusing on the most urgent hazards. We propose an approach that translates psychosocial risk questionnaire scores into concrete risks for health impairment. We believe that risk values correspond much more to lived reality and are easier to understand than the coefficients commonly used in work and organizational psychology, such as correlations or regression weights. Thus, we established an approach that enables even laypersons to engage in informed (political) discussions on the potential impact of certain job stressors and to decide whether a measured expression of a job stressor is acceptable or not. For example, the different health and safety actors of an organization could define a risk of 30% for psychological health impairment (compared to a 15% average risk in the population) as unacceptable and a mandatory threshold for work redesign measures to reduce psychological hazards. With the results of logistic regression analysis, this risk value can be backtranslated into a questionnaire score for each psychological job stressor and job resource that constitutes the cutoff value for this specific questionnaire scale.

Furthermore, our proposed method accounts for the central findings of work stress research because it incorporates the idea that both individual stressors and combinations of job stressors and job resources may affect employee health. Thus, we can use the results of multiple logistic regressions analyses to calculate risk values for different combinations. Depending on the expression of matching job resources, this may result in differentiated cutoff values for job stressors [41].

### 4.3. Limitations and Further Research Requirements

The proposed method for translating questionnaire results into risk values has methodological limitations to be accounted for. A methodological criticism may concern the dichotomization of the outcome variable psychological health impairment. The artificial dichotomization and transformation of continuous variables into dichotomous variables is accompanied by a loss of information that can be criticized because the result offers less differentiation. For instance, we merged values of 5, 6, and 7 and contrasted them with the merged values of 1–4 on the emotional exhaustion scale. However, this disadvantage is more than compensated for by the advantage of a clear and understandable presentation of results that may increase the impact of the results. For this reason, dichotomization is very common in epidemiology, medicine, and some areas of psychology [42]. Many allegedly naturally dichotomous variables are artificially dichotomized, such as adiposity as a dichotomization of body mass index or diabetes as a dichotomization of continuous HbA1c (Glycated hemoglobin) scores [43,44]. With emotional exhaustion, we considered different references that defined severe burnout [34] or high exhaustion based on statistical deviation from a norm [36]. However, we decided to rely on precise and transparent frequency statements (“weekly symptoms”) that are considered to represent a severe health impairment judged by practitioners. To limit the arbitrary nature of expert judgements, more research is needed to define threshold values for psychological health measures, not only for emotional exhaustion.

Another limitation of the presented approach may be the selection of the outcome variable itself: emotional exhaustion. The effects of work stressors on employee health may be unspecific and diffuse. Therefore, we assume that problematic work stress does not necessarily manifest or result only in emotional exhaustion. Work psychology and stress research has shown that the detrimental effects of work stress may manifest in a variety of health impairments, including somatic complaints [4]. We suggest applying the proposed method to a variety of different health outcomes and incorporating combinations of symptoms of different areas.

Another requirement for further research concerns the cross-sectional nature of the dataset used in our data example. Health and safety approaches, in general, and psychosocial risk assessments, in particular, follow a preventive logic. Therefore, health risk assessment should adopt a prospective and prognostic perspective and, thus, should do the analysis. Future research should investigate the predicted values in comparison to actual health impairments by conducting longitudinal studies.

Finally, with regard to ecological validity, our data only accounts for individual data. We predict and calculate risks for individual psychological health impairment based on individually assessed job stressors and job resources. Therefore, we use the questionnaire scores on the respective job stressors and job resources scores of individuals. In the practical application of psychosocial risk assessment, however, individual data are rarely available. Instead, the psychological risk is assessed based on aggregated data of incumbents in similar workplaces. Thus, future studies should strive to replicate our methodological approach with aggregated data for job stressors, job resources, and health outcomes.

## Figures and Tables

**Figure 1 ijerph-18-07107-f001:**
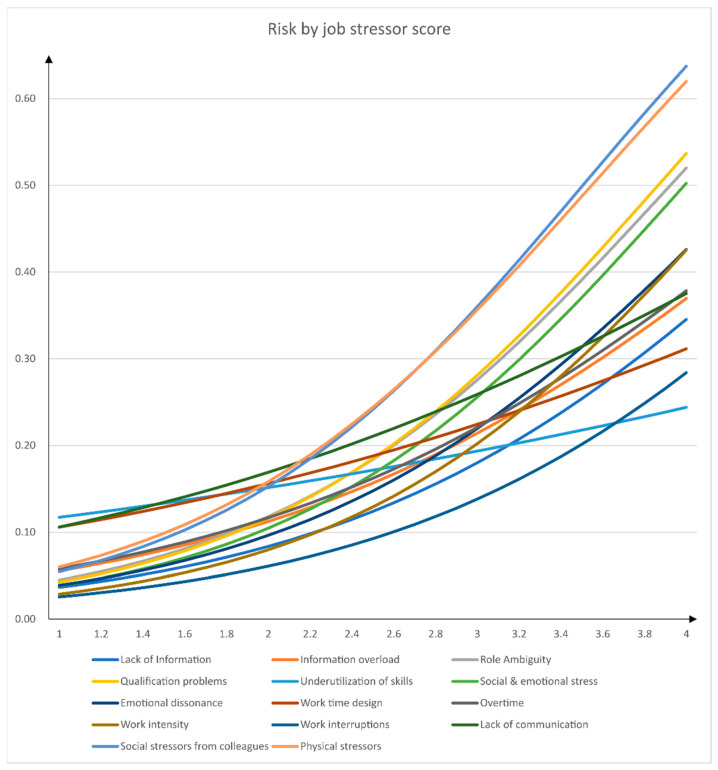
Relationship between job stressors and risk for psychological health impairment.

**Figure 2 ijerph-18-07107-f002:**
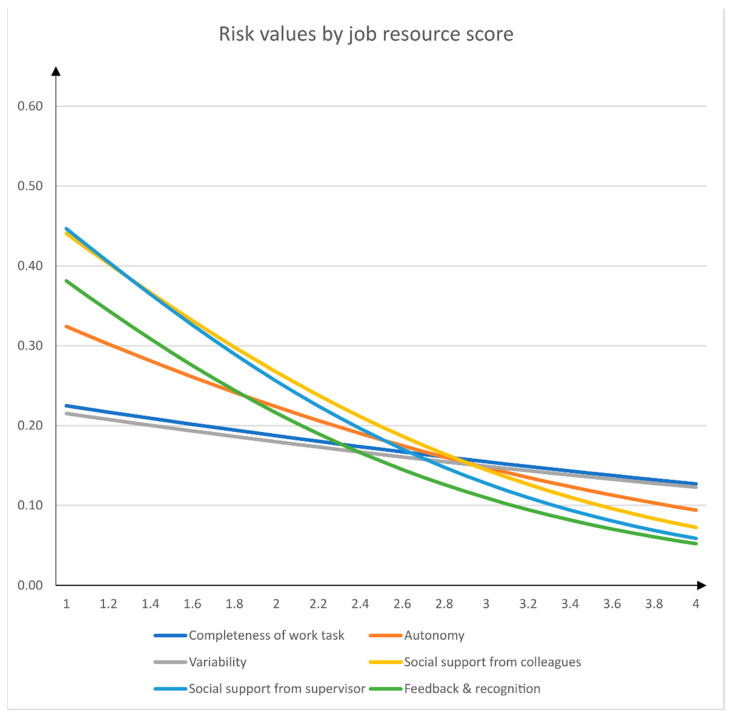
Relationship between job resources and risk for psychological health impairment.

**Table 1 ijerph-18-07107-t001:** Overview of FGBU: scales, sample items, Cronbach’s alphas, means, and standard deviations.

Scale	Sample Item	*α*	*M* (*SD*)
**Work task content**			
Completeness of work task (CWT)	At work, I am involved in work processes from start to finish.	0.71	3.11 (0.71)
Autonomy (AUT)	I have a lot of freedom in the way I do my job.	0.84	3.02 (0.74)
Variability (VAR)	My work tasks change frequently.	0.77	2.90 (0.66)
Lack of information (INFO_L)	The lack of necessary information hampers decision-making.	0.83	2.61 (0.76)
Information overload (INFO_O)	The daily amount of information is too much (e.g., mails, corporate communications).	0.85	2.30 (0.80)
**Responsibility and skills**			
Role ambiguity (ROLE_A)	The priorities between different work goals are not clearly defined.	0.77	2.08 (0.76)
Qualification problems (QP)	I often feel overwhelmed by the type of tasks I am given.	0.81	2.06 (0.79)
Underutilization of skills (UU)	My work is not challenging.	0.83	1.97 (0.79)
**Emotional load**			
Social and emotional stress (SO_EM)	My work sometimes causes states that are very emotionally draining (e.g., grief, rage).	0.83	2.15 (0.84)
Emotional dissonance (E_DISS)	At work, I often have to hide my real feelings.	0.90	2.29 (0.88)
**Work organization**			
Work time design (WTD)	I have highly variable working hours.	0.82	1.83 (0.85)
Overtime (OT)	I often work overtime.	0.81	2.21 (0.86)
Work intensity (INT)	The high volume of work often causes intense time pressure.	0.89	2.40 (0.91)
Work interruptions (WI)	During my work I am often interrupted by other people.	0.88	2.94 (0.81)
Lack of communication (LOC)	My workplace lacks opportunities for personal interaction.	0.84	1.67 (0.76)
**Social relations**			
Social support from colleagues (SUPP_C)	I can talk openly about everything with my colleagues.	0.88	3.05 (0.73)
Social stressors from colleagues (STRESS_C)	There are often tensions between colleagues.	0.84	1.80 (0.72)
Social support from supervisors (SUPP_S)	My supervisor is ready to listen to my problems.	0.90	2.95 (0.81)
Feedback and recognition (F and R)	I am recognized for my work.	0.87	2.71 (0.82)
**Work environment**			
Physical stressors (PHY)	Poor ergonomic design.	0.89	1.80 (0.66)

*Note*. CWT = Completeness of work task; AUT = Autonomy; VAR = Variability; INFO_L = Lack of information; INFO_O = Information overload; Role_A = Role ambiguity; QP = Qualification problems; UU = Underutilization of skills; SO_EM = Social and emotional stress; E_DISS = emotional dissonance; WTD = Work time design; OT = Overtime; INT = Work intensity; WI = Work interruptions; LOC = Lack of communication; SUPP_C = Social support from colleagues; STRESS_C = Social stressors from colleagues; SUPP_S = Social support from supervisors; F and R = Feedback and recognition; PHY = Physical stressors.

**Table 2 ijerph-18-07107-t002:** Intercorrelations for all study variables.

Variables	1	2	3	4	5	6	7	8	9	10	11	12	13	14	15	16	17	18	19	20
1 EE	1.0 **																			
2 CWT	−0.07 **																			
3 AUT	−0.20 **	0.35 **																		
4 VAR	−0.07 **	0.37 **	0.30 **																	
5 INFO_L	0.33 **	0.09 **	0.02	0.24 **																
6 INFO_O	0.33 **	0.05 **	−0.04 **	0.20 **	0.57 **															
7 ROLE_A	0.39 **	−0.09 **	−0.12 **	0.03	0.48 **	0.44 **														
8 QP	0.42 **	−0.04 **	−0.08 **	0.07 **	0.45 **	0.44 **	0.58 **													
9 UU	0.12 **	−0.23 **	−0.09 **	−0.29 **	0.07 **	0.03	0.31 **	0.15 **												
10 SO_EM	0.43 **	−0.01	−0.19 **	0.07 **	0.29 **	0.28 **	0.38 **	0.37 **	0.11 **											
11 E_DISS	0.42 **	−0.04 **	−0.19 **	0.03	0.27 **	0.26 **	0.36 **	0.36 **	0.16 **	0.65 **										
12 WTD	0.20 **	−0.00	−0.20 **	0.10 **	0.16 **	0.21 **	0.24 **	0.22 **	0.08 **	0.42 **	0.33 **									
13 OT	0.35 **	0.13 **	−0.07 **	0.24 **	0.34 **	0.40 **	0.30 **	0.36 **	−0.07 **	0.39 **	0.35 **	0.53 **								
14 INT	0.48 **	0.12 **	−0.11 **	0.18 **	0.40 **	0.48 **	0.34 **	0.41 **	−0.11 **	0.40 **	0.35 **	0.35 **	0.71 **							
15 WI	0.37 **	0.10 **	−0.03 *	0.21 **	0.41 **	0.39 **	0.30 **	0.34 **	−0.04 **	0.32 **	0.31 **	0.16 **	0.42 **	0.54 **						
16 LOC	0.22 **	−0.09 **	−0.08 **	−0.05 **	0.16 **	0.21 **	0.30 **	0.28 **	0.23 **	0.24 **	0.24 **	0.30 **	0.23 **	0.21 **	0.05 **					
17 SUPP_C	−0.29 **	0.16 **	0.19 **	0.19 **	−0.13 **	−0.13 **	−0.25 **	−0.21 **	−0.17 **	−0.16 **	−0.21 **	−0.10 **	−0.12 **	−0.16 **	−0.05 **	−0.28 **				
18 STRESS_C	0.40 **	−0.10 **	−0.15 **	−0.07 **	0.25 **	0.25 **	0.44 **	0.38 **	0.27 **	0.44 **	0.38 **	0.29 **	0.29 **	0.30 **	0.20 **	0.35 **	−0.45 **			
19 SUPP_S	−0.35 **	0.16 **	0.18 **	0.17 **	−0.17 **	−0.14 **	−0.34 **	−0.27 **	−0.18 **	−0.23 **	−0.21 **	−0.10 **	−0.15 **	−0.20 **	−0.11 **	−0.18 **	0.50 **	−0.35 **		
20 F and R	−0.33 **	0.18 **	0.19 **	0.19 **	−0.15 **	−0.10 **	−0.30 **	−0.23 **	−0.21 **	−0.19 **	−0.20 **	−0.06 **	−0.07 **	−0.14 **	−0.13 **	−0.12 **	0.43 **	−0.28 **	0.68 **	
21 PHY	0.37 **	−0.06 **	−0.28 **	−0.04 **	0.20 **	0.23 **	0.32 **	0.29 **	0.23 **	0.41 **	0.34 **	0.48 **	0.34 **	0.31 **	0.19 **	0.30 **	−0.19 **	0.41 **	−0.22 **	−0.20 **

*Note*. *N* = 3336 − 4210; ** = *p* < 0.01; * = *p* < 0.05;; EE = emotional exhaustion; CWT = completeness of work; AUT = autonomy; VAR = variability; INFO_L = lack of information; INFO_O = information overload; ROLE_A = role ambiguity; QP = qualification problems; UU = underutilization of skills; SO_EM = social and emotional stress; E_DISS = emotional dissonance; WTD = work time design; OT = overtime; INT = work intensity; WI = work interruptions; LOC = lack of communication; SUPP_C = social support from colleagues; STRESS_C = social stressors from colleagues; SUPP_S = social support from supervisors; F and R = feedback and recognition; PHY = physical stressors.

**Table 3 ijerph-18-07107-t003:** Prediction of psychological health impairment based on single job stressor and resources scores.

Predictors	Intercept	*B*	Probability for Psychological Health Impairment for Questionnaire Score			
			1	2	3	4
**Job Stressors**						
Lack of information (INFO_L)	−4.15	0.88	0.04	0.08	0.18	0.35
Information overload (INFO_O)	−3.60	0.77	0.06	0.11	0.21	0.37
Role ambiguity (ROLE_A)	−4.10	1.05	0.04	0.12	0.28	0.52
Qualification problems (QP)	−4.20	1.09	0.04	0.12	0.28	0.54
Underutilization of skills (UU)	−2.32	0.30	0.12	0.15	0.19	0.24
Social and emotional stress (SO_EM)	−4.30	1.08	0.04	0.10	0.26	0.50
Emotional dissonance (E_DISS)	−4.18	0.97	0.04	0.10	0.22	0.43
Work time design (WTD)	−2.58	0.45	0.11	0.16	0.22	0.31
Overtime (OT)	−3.55	0.76	0.06	0.12	0.22	0.38
Work intensity (INT)	−4.59	1.07	0.03	0.08	0.20	0.43
Work interruptions (WI)	−4.55	0.91	0.03	0.06	0.14	0.28
Lack of communication (LOC)	−2.67	0.54	0.11	0.17	0.26	0.38
Social stressors from colleagues (STRESS_C)	−3.99	1.14	0.05	0.15	0.36	0.64
Physical stressors (PHY)	−3.83	1.08	0.06	0.16	0.36	0.62
**Job Resources**						
Completeness of work task (CWT)	−1.01	−0.23	0.22	0.19	0.15	0.13
Autonomy (AUT)	−0.23	−0.51	0.32	0.22	0.15	0.09
Variability (VAR)	−1.07	−0.22	0.22	0.18	0.15	0.12
Social support from colleagues (SUPP_C)	0.53	−0.77	0.44	0.27	0.14	0.07
Social support from supervisor (SUPP_S)	0.64	−0.85	0.45	0.26	0.13	0.06
Feedback and recognition (F and R)	0.32	−0.81	0.38	0.22	0.11	0.05

*Note*. CWT = Completeness of work task; AUT = Autonomy; VAR = Variability; INFO_L = Lack of information; INFO_O = Information overload; Role_A = Role ambiguity; QP = Qualification problems; UU = Underutilization of skills; SO_EM = Social and emotional stress; E_DISS = emotional dissonance; WTD = Work time design; OT = Overtime; INT = Work intensity; WI = Work interruptions; LOC = Lack of communication; SUPP_C = Social support from colleagues; STRESS_C = Social stressors from colleagues; SUPP_S = Social support from supervisors; F and R = Feedback and recognition; PHY = Physical stressors.

**Table 4 ijerph-18-07107-t004:** Scale score combinations and risks for psychological health impairments.

Job Stressors	Scale Score	Autonomy			
		1	2	3	4
**Work intensity**	1	0.07	0.04	0.03	0.02
	2	0.16	0.12	0.08	0.05
	3	0.37	0.28	0.19	0.13
	4	0.63	0.52	0.41	0.31
**Social support from supervisor**					
	Scale score	1	2	3	4
**Social and emotional stress**	1	0.13	0.07	0.04	0.02
	2	0.24	0.17	0.09	0.05
	3	0.53	0.36	0.21	0.12
	4	0.76	0.60	0.43	0.27

*Note.* Regression coefficients for work Intensity/autonomy-model = −3.22 + score work intensity scale × 1.05 + score autonomy scale × −0.45. Regression coefficients for social and emotional stress/social support from supervisor-model = −2.14 + score social and emotional stress scale × 1.00 + score social support scale × −0.72. × = multiplication sign.

**Table 5 ijerph-18-07107-t005:** Cross tables and percent correct classifications (PCC).

Scale	Score	Psychological Health Impairment		PCC
		0	1	
Lack of information ≥ 3.8	0	3373	526	82.92%
	1	193	117	
Information overload ≥ 3.6	0	3359	525	82.59%
	1	208	118	
Role ambiguity ≥ 3.1	0	3351	479	83.49%
	1	216	164	
Qualification problems ≥ 3.1	0	3322	477	82.85%
	1	245	166	
Social and emotional stress ≥ 3.2	0	3232	429	81.85%
	1	335	214	
Emotional dissonance ≥ 3.5	0	3314	471	82.80%
	1	253	172	
Work time design ≥ 3.9	0	3478	602	83.59%
	1	89	41	
Overtime ≥ 3.6	0	3349	502	82.90%
	1	218	141	
Work intensity ≥ 3.5	0	3253	411	82.80%
	1	313	232	
Lack of communication ≥ 3.4	0	3508	595	84.47%
	1	59	48	
Social stressors from colleagues ≥ 2.8	0	3330	434	84.08%
	1	236	209	
Physical stressors ≥ 2.8	0	2638	382	82.79%
	1	193	129	
Autonomy ≤ 1.2	0	3505	612	83.99%
	1	62	31	
Social support from colleagues ≤ 1.8	0	3418	554	83.32%
	1	148	89	
Social support from supervisor ≤ 1.7	0	3328	482	83.03%
	1	231	160	
Feedback and recognition ≤ 1.4	0	3323	500	82.39%
	1	241	143	

## Data Availability

Data available on request from the corresponding author.

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
