# Peer review of "How to Use Questionnaire Results in Psychosocial Risk Assessment: Calculating Risks for Health Impairment in Psychosocial Work Risk Assessment"

_ijerph, 2021, doi:10.3390/ijerph18137107_

Round 1

Reviewer 1 Report

This manuscript proposes a novel way of solving existing problems of using means for cut off scores of psychosocial measures. The use of risk values instead of absolute scores cater individual differences and is therefore a better and viable advancements.

Specific comments:

  1. Please remove 'is' before 'required' in line 15, p. 3.
  2. Please consider simplifying the following sentence as it contains too many ideas: ' As job stressors act mainly as a chronic impact of the work situation (e.g., daily hassles) rather than as single events, Clark and Cooper [21] consider the chronic intensity of a psychological stressor (exposure) as it is measured by a questionnaire as the probability in their risk definition.' (lines 10 -12, first paragraph, section 1.6)
  3. The meaning of 'cross tables' in the following sentence needs not clarification: '... we calculated cross tables for low and high scores of the job stressor and job resources scales and psychological health impairment.' (p. 13, line 3 of the paragraph below Table 4)
  4. Please correct the typo in the following part: ' and health out-comesReferences must be numbered in order of appearance' (p. 16, last paragraph, lines 8 - 9).

Author Response

Thank you very much for your comments.

Please see the attachment for my reply.

Reviewer 2 Report

Due May 31, 2021

Review request: International Journal of Environmental Research and Public Health

Manuscript ID: ijerph-1223895

Title:    How to use questionnaire results in psychosocial risk assessment-Calculating risks for health impairment in psychosocial work risk assessment

Synopsis

This methodology paper proposes an empirically derived method of interpreting the level of risk of psychosocial health impairment from the Psychosocial Work Risk Assessment (FGBU). The psychosocial health impairment was assessed by a single question of “I feel burned out from my job” and rated by a 7-point scale. Table 1 summarizes the survey questionnaire items. Table 2 is a correlation matrix between the survey items including the response variable. Table 3 reports the regression coefficient of individual predictors by four different probabilities of psychological health impairment. Figure 1 shows the relations between the psychological health risk value and single stress score, indicating a positive relationship for stressors and a negative relationship for resources. Table 4 shows the attempt to combine the negative and positive predictors. Table 5 summarizes crosstabs of dichotomously divided factors and psychological health impairment.   

Reviewer's conflict of interest: None

General comment to authors

The manuscript should have line numbers in order to locate the comment of discussion. I hope authors are able to identify the specific locations regarding to my comments. Grammatical or typographical errors need to be reviewed more thoroughly. Overall, a figure and some tables can use detailed captions. Logistic regression analyzes multiple correlations. I order to analyze pathways, this study may be able to construct a structural equation model and determine the fitness of the model; however, the authors’ attempt to construct a regression model of interpretation was interesting.

Specific comments are listed below:

  1. Title: a good title.
  2. Abstract: Well written. “e.” is missing between “the burnout indicator” and “emotional exhaustion.”
  3. Introduction:
    1. The first sentence in 1.1. To preserve employee’s health and prevent accidents…
    2. The second sentence in 1.2. …often structured and differentiated between…
    3. The fourth sentence in 1.2. Delete “is” prior to “required is not trivial.”
    4. The sixth sentence in 1.2. This sentence may be clearer: “the survey provides confirmatory results that have to be integrated…”
    5. Section 1.4, the fourth paragraph. The ASCII characters for the standard deviation of 0.5 seem to be offset.

  1. Methods and Materials:
  2. 2.1. “The questionnaire consist of 19 3-items scales and an index of 10 physical stressors.” Table 1 shows 20 items. Please clarify the total number of FGBU questionnaire items.
  3. 3. Analysis. The last sentence. In these cases, we calculated multiple logistic regression analyses entering both…” Change analyses to “equations.”

  1. Results:
    1. Table 3. To be consistent, the stressors and resources can use the abbreviation in parentheses.
    2. The heading, “Job Stressors” should be written below the line. The heading of the first column should be “Predictors” which indicate both stressors and resources.
    3. Figure 1. Please add the labels for the y- and x-axes.
  2. Discussion:
    1. No comment.
  3. Author’s Declaration such as funding, author’s contribution, and conflict of interest statements may be missing.

End of review.

Author Response

(The authors gave the same response as above.)

Reviewer 3 Report

I appreciate the opportunity to review this manuscript titled How to Use Questionnaire Results in Psychosocial Risk Assessment - Calculating Risks for Health Impairment in Psychosocial Work Risk Assessment.

The main issue raised is of great interest. Offering a method of orientation and guidance to practitioners on how to interpret questionnaires results that are applied in the risk assessment process can be highly useful.  The authors develop a correct approach to the problem and justification of the investigation. The literature review is adapted to the subject of the study.

The objective of the study is well defined and responds to the problem posed. In addition, its subsequent execution through the development of the research meets all the scientific and research requirements

The authors correctly justify the application of logistic regression as an analysis, since this type of statistical test aims at predicting the probability of a dichotomous outcome variable. I believe that the argument that they indicate to use this type of analysis and not linear regression is correct and therefore its application responds to the proposal of the manuscript

The instrument applied (FGBU; Questionnaire for Psychosocial Work Risk Assessment) is correct and is also based on the recommendations of the Joint German Occupational Health and Safety Strategy; therefore, its application is justified. Also the Maslach Burnout Inventory (MBI), is widely used in psychosocial risks research, the authors offer all the necessary information of this instrument (version, Cronbach’s alpha, ...)

The final number of participants is a strength to highlight in the study.

Although, as the authors point out, the abbreviations are already in Table 1, it could be interesting to also include them in Table 2 of correlations as a note. In addition, the correlation table seems difficult to interpret in the current format, as a recommendation I believe that it is better not to use bold, instead they could put * for p <.01 and ** for p <.05.

The results provide all the information necessary to understand the procedure and to be able to replicate or apply the same process.

The discussion is consistent with the results and the practical implications are highlighted.

Author Response

(The authors gave the same response as above.)
